# Alienation in Ethiopian cinema: *"T'eza" ("Morning Dew")* and *"Səlä ʾänəči" ("About you")* in focus

**Kindie Abebaw**◉*, Anteneh Aweke, Sintayehu Genet

Department of English Language and Literature, Faculty of Humanities, Bahir Dar University, Bahir Dar, Ethiopia

\* kinabebaw82@gmail.com

## Abstract

The main objective of this study was to show and analyze the representation of alienation in selected Amharic films. The films selected for the analysis include *T'eza (Morning Dew)* (2008) by Haile Gerima and *Səlä ʾänəči (About you)* (2009) by Belay Getaneh. The study explores how the selected Amharic films depict characters facing estrangement and marginalization in Ethiopian society, focusing on their struggles with alienation and identity. It aims to uncover the psychological and social dynamics behind these experiences and provide insights into themes of identity, loss, and societal exclusion. Using psychoanalytic and Marxist theories, the study analyzed the psychological, cultural, political, and social dimensions of alienation portrayed in these films. This study is sought with an interpretative paradigm and qualitative approach, employing narrative and descriptive designs. The films serve as the primary source of data, supplemented by secondary data from journals, books, and internet sources to provide additional context and theoretical support. The study highlights varied experiences of alienation among main characters, aiming to deepen understanding of this theme in Ethiopian cinema and inform efforts to address alienation in Ethiopian society. The significance of the study lies in its contribution to understanding this theme within the Ethiopian context, specifically through the lens of Amharic films. In conclusion, the study's findings and analysis can inform discussions and interventions aimed at addressing alienation and its consequences in Ethiopian society.

## 1. Introduction

Literature reflects and shapes the human condition by conveying diverse ideas, emotions, and narratives. It provides insights into human complexities across cultures and history, influencing and mirroring societal norms and changes. Literature powerfully expresses ideas, emotions, and experiences, offering an imaginative reflection of human life that both mirrors and transcends reality [26]. The study examines the relationship between literature and psychoanalytic as well as Marxist theories. By analyzing alienation in films through these frameworks, the study deepens understanding of the films' themes. Psychoanalysis views literature as a mirror of our inner desires and mental states [31] and Marxism also examines literature's connection to the socio-political aspects of its time. Literature and film serve as

**Data availability statement:** The data is available in the manuscript it self.

**Funding:** The author(s) received no specific funding for this work.

**Competing interests:** The authors have declared that no competing interests exist.

powerful mediums for expressing ideas, emotions, and experiences. Film is an artistic medium that blends visual, imagery, audio, and narrative elements to tell stories and convey ideas [12]. The film evolves with new techniques and genres, dynamically interacting with and drawing inspiration from literature [24].

Despite Ethiopia's ancient literary tradition, its film industry is relatively young, offering unique insights into the nation's societal struggles [7]. Amharic films, seen as extensions of literature, foster a shared cultural experience. Following the political shift in the 1990s, the industry experienced a boom with a surge in independent productions and international recognition, though its output remains modest compared to Ethiopia's rich cultural heritage [25]. The Ethiopian film industry has long been recognized for its distinct storytelling approaches and strong cultural influences. According to [23], Ethiopian films offer a window into the nation's rich history and diverse traditions, revealing unique insights into its cultural essence. Similarly [8], highlights the way these films convey distinct perspectives that are deeply rooted in the Ethiopian experience. Ethiopian filmmakers have established a distinctive presence in global cinema by sharing narratives that reflect the nation's rich cultural identity. These unique stories provide a compelling alternative to mainstream film traditions, gaining recognition from both audiences and scholars.

Amharic films capture Ethiopia's sociocultural realities, portraying its struggles and joys while driving social change. Despite challenges, they enhance the understanding of Ethiopian identity. This study offers a fresh perspective by examining the overlooked theme of alienation in Amharic cinema through psychoanalytic and Marxist theories. By doing so, it fills a scholarly gap and provides new insights into how these films reflect societal realities, contributing valuable knowledge to the field of Ethiopian cinema in comparison with other African cinemas.

In examining the phenomenon of alienation in Amharic films, particularly in the context of *T'eza (Morning Dew)* and *Sǝlä ʾänǝči (About You)*, it is essential to acknowledge the foundational work of previous studies that have significantly contributed to the understanding of Ethiopian cinema. Notable scholars such as [3] and [37] have explored the development and unique characteristics of Ethiopian films, situating them within the broader narrative of African cinema and the local film industry. Their explorations provide vital insights into the cultural, aesthetic, and socio-political factors that inform Amharic filmmaking, allowing for a more comprehensive appreciation of the cinematic landscape in Ethiopia. While their studies provide valuable insights into the development of Ethiopian cinema, they largely focus on the industry's broader characteristics, often neglecting the psychological and existential dimensions of alienation in films. Our research fills this gap by applying psychoanalytic and Marxist frameworks to explore how Amharic films, particularly *T'eza (Morning Dew)* and *Sǝlä ʾänǝči (About You)*, depict various forms of alienation—psychological, cultural, political, social, economic and existential—reflecting deeper societal issues within Ethiopian culture.

[15] work, 'A Critical Discourse Analysis of Ethiopian Amharic Movie Entitled '*T'eza*," employs critical discourse analysis (CDA) to investigate the socio-political power dynamics and social hierarchies expressed through the film's dialogue. However, Alienation in Ethiopian Cinema extends this analysis by applying psychoanalytic and Marxist frameworks to explore the deeper layers of psychological, cultural, political, social, economic, and existential alienation. By examining both films, this study provides a more holistic understanding of alienation in Ethiopian cinema, addressing not only external socio-political forces but also internal struggles with identity and cultural displacement, thereby offering a subtle exploration of both societal and personal dimensions of alienation.

The Black Camera special issue analyzes Haile Gerima's *T'eza* through Frantz Fanon's theories, blending Marxist and psychoanalytic perspectives on colonialism, identity, and

alienation, with [36] providing an annotated bibliography on Gerima's work. While Black Camera situates *T'eza* in broader postcolonial contexts, our study fills a gap by focusing on internal forms of alienation—psychological, cultural, political, social, economic, and existential—within Ethiopian society. By analyzing *Səlä ʾänəči* alongside *T'eza*, we expand the discourse on alienation in Ethiopian cinema, offering a more nuanced understanding of these themes.

[11] work, 'Revolutionary Ethiopia through the Lens of the Contemporary Film Industry', uses psychoanalytic methods to analyze Ethiopian films, including *T'eza*, focusing on the psychological effects of revolution and societal transformation. While her study centers on the impact of political disruption, our research expands the psychoanalytic framework to explore broader themes of psychological, cultural, political, social, economic, and existential alienation in Ethiopian society. Unlike [11] revolution-centered analysis, our study explores deeper into existential and personal alienation, particularly in the context of cultural dislocation and internal societal tensions, offering a more comprehensive exploration of alienation.

This focus contrasts with the more global critique of alienation, offering new insights into the psychological processes at play in post-revolutionary Ethiopian society. [11] study examines *T'eza* through Frantz Fanon's postcolonial theory, emphasizing colonialism, cultural alienation, and racial identity, focusing on the psychological effects of colonialism and identity reconstruction through resistance. In contrast, our study combines psychoanalytic and Marxist lenses to explore internal and localized alienation. We analyze Anberber's political disillusionment and Azanu's social rejection in *T'eza* and familial trauma and emotional detachment in *Səlä ʾänəči*, highlighting psychological and existential alienation.

Similarly [39], examines Ethiopian films through Jameson's Marxist critique of global capitalism, focusing on external forces like Westernization and cultural commodification, our study shifts to internal societal alienation and introduces a psychoanalytic perspective to uncover characters' inner struggles and existential crises. [39] article, applying Fredric Jameson's Marxist theories, examines how Westernization shapes Ethiopian films like *Rebuni* and *YeWendoch Guday*, highlighting the tension between cultural preservation and global market pressures. He argues that these films undergo cultural commodification, adapting local narratives to Western frameworks. In contrast, our study shifts the focus to internal forms of alienation— psychological, cultural, political, social, economic, and existential —within Ethiopia. For instance, *Səlä ʾänəči (About You)* explores class conflict and the father's internal struggles, while *T'eza (Morning Dew)* examines Anberber's personal and cultural dislocation.

The 2018 collection Cine-Ethiopia: The History of Film and Politics in the Horn of Africa analyzes the socio-political evolution of Ethiopian cinema, focusing on how filmmakers address historical and contemporary issues [38]. While it provides a broad political perspective, our study addresses a gap by concentrating on themes of alienation, offering deeper insights into personal and societal dislocation, psychological trauma, and identity struggles. By integrating the insights from Cine-Ethiopia, we link these personal narratives to the broader socio-political context, enriching our analysis and providing a more subtle understanding of alienation in Ethiopian cinema.

Therefore, this study builds on prior research to examine the psychological alienation originating from the Derg regime, focusing on the internal struggles of characters like Anberber in *T'eza*. Drawing on works by [3] and [11], we explore how political repression and cultural dislocation intersect with personal trauma to deepen estrangement. Unlike others global perspectives, our localized psychoanalytic approach highlights the personal, emotional, and cultural disintegration faced by individuals in post-revolutionary Ethiopian society, offering new insights into the legacy of historical oppression. In comparison to studies like [39] Marxist Analysis in *Black Camera*, which focuses on the global impacts of capitalism on Ethiopian

cinema, our study provides a localized psychoanalytic examination of the internal, psychological alienation faced by individuals under the Derg. Our work focuses on the personal and emotional dimensions of alienation, exploring how characters in *T'eza* and *Səlä 'änəči* contend with feelings of loss, cultural disintegration, and political disillusionment—experiences that are rooted in the Derg era's oppressive environment.

In general, this study employs psychoanalytic and Marxist theories to examine alienation in Amharic films, focusing on *T'eza* and *Səlä 'änəči*. Addressing gaps in Ethiopian cinema study, it analyzes alienation on individual and societal levels, exploring its psychological, cultural, political, social, economic, and existential dimensions. Highlighting the Derg regime's oppressive socio-political conditions (1974–1991), the study examines how authoritarianism, civil war, and repression caused trauma, dislocation, and ideological crises, particularly among intellectuals and artists. By centering localized struggles rather than global capitalist critiques, this study illuminates the emotional impact of alienation within Ethiopia's unique political history, enriching global cinema studies.

For further detail, the following research questions are formulated:

1. How do the selected Amharic films depict the impact of alienation on characters?

2. How are the different forms of alienation represented in selected films?

## 2. Theoretical framework

### 2.1. Psychoanalysis

In the films, Freud's and Lacan's psychoanalytic theories offer critical insights into alienation. Freud emphasizes that repressed desires shape unconscious behavior, often leading to self-alienation through mechanisms like repression [17]. Lacan expands this, arguing that desires are structured by societal symbols, creating a pervasive sense of lack and alienation. He distinguishes between the 'Imaginary' self and the 'Real' chaotic truths that disrupt this idealized self-image, intensifying feelings of alienation [27]. These frameworks highlight the deep psychological and social alienation the characters experience in navigating societal expectations.

Applying psychoanalytic frameworks to *Səlä 'änəči* and *T'eza* uncovers deeper layers of the character's motivations and experiences, revealing how unconscious emotional states shape their struggles with identity and belonging. Psychoanalytic criticism highlights the authors' embedded personal and societal alienation [4,40]. This approach offers a deeper understanding of the psychological and cultural forces driving the characters' alienation within contemporary Ethiopian society. Psychoanalytic criticism, based on Freud's theories, explores the psychological conflicts in literature, revealing emotional complexities within texts. [9] emphasizes that this approach uncovers emotional depth, especially in works portraying intricate emotional struggles. Freud's concepts of the '*id*,' '*ego*,' and '*superego*' illustrate the internal conflicts in human experience, with the *ego* mediating between primal desires and moral imperatives, often resulting in alienation [18,19]. This framework provides insight into the psychological dimensions of alienation in literature.

[21] argues that societal structures shape emotions and anxieties, leading to alienation as a disconnection from oneself and others. [17] links alienation to repression, where repressed desires manifest in disguised forms, creating an 'ego-alien' split. This internal estrangement causes psychological distress as these repressed conflicts appear foreign and threatening to the conscious mind. This framework highlights how sociocultural factors contribute to personal alienation and psychological turmoil. [20] argues that modern industrial society, driven by consumerism and powerlessness, deepens feelings of alienation. Applying psychoanalytic

concepts to *Sələ ʾänəči* and *Tʾeza* reveals how the characters confront internal conflicts and societal pressures. This approach offers a refined understanding of their psychological distress and alienation within the broader context of Ethiopian society. In conclusion, *Tʾeza* and *Sələ ʾänəči* depict various forms of alienation—psychological, emotional, existential, and cultural—using psychoanalytic lenses to explore the characters' inner struggles and disconnection from cultural norms.

## 2.2.  Marxist criticism

Marxist literary theory argues that literature reflects the social and material conditions of its time, shaped by historical forces of production and exchange [16]. Marx highlights the class struggle between the bourgeoisie and proletariat as central to understanding societal structures, with literary criticism often examining how this conflict is portrayed [5]. This lens reveals the interplay between culture, politics, and social alienation within texts. A Marxist approach to literature reveals power imbalances, especially in class relations, by analyzing texts as products of their historical and economic contexts, rather than timeless art [5,41]. This perspective highlights how societal structures influence narratives of cultural, political, and social alienation.

Using a Marxist lens to analyze *Sələ ʾänəči* and *Tʾeza* highlights the political, cultural, and social alienation of characters shaped by oppressive societal structures. The films portray how class dynamics and socio-economic disparities deepen the characters' disconnection from their heritage and community, revealing the complex interplay of individual struggles and systemic oppression in Ethiopian society. This approach emphasizes the relevance of Marxist theory in understanding these films' exploration of alienation and identity [5,41].

Marxist literary criticism views literature as a reflection of the socio-economic and ideological dynamics of its time [5]. By examining social inequality and power structures, this approach reveals how literature functions as a social institution shaped by its historical context rather than universal standards [41]. Marx's concept of alienation, a result of capitalist structures, shows how workers are disconnected from their labor, themselves, and society [28,35]. [22] and [6] further explore how ideology and social institutions perpetuate alienation under capitalism.

Analyzing *Sələ ʾänəči* and *Tʾeza* through a Marxist lens reveals how characters' cultural, political and social alienation stems from socio-economic conditions shaped by capitalist structures. Marx's concept of alienation argues that workers are estranged from their labor and humanity due to a loss of control over production [30,33]. In these films, characters experience disconnection from their labor, identities, and communities, reflecting social inequality and class conflict in Ethiopian society. Marxist theory underscores how oppressive socio-political systems deepen personal and cultural alienation, offering a holistic view of these struggles [29]. In conclusion, Marxist theory reveals how socio-political forces intensify this estrangement, offering a comprehensive understanding of both personal and systemic influences on alienation in these films.

## 3.  Methodology of the study

The study uses an interpretivist approach to understand how Amharic films portray alienation. This makes it suitable for analyzing how Amharic films depict alienation, considering the subjective experiences and diverse realities within Ethiopian society through the researchers' experiences, knowledge, and social-cultural interactions. The study employs a qualitative approach, focusing on content and textual analysis as the primary means of investigating the depiction of alienation in Amharic films. The primary sources for the study are the selected

Amharic films, supplemented with secondary sources. The selected films are *T'eza (Morning Dew)* (2008) directed by Haile Gerima and *Səlä ʾänəči (About you)* (2009) directed by Belay Getaneh.

We employed purposive sampling to select films that prominently depict alienation, focusing on genre, character actions, content, accessibility, and the directors' backgrounds. The selected films were watched repeatedly to analyze their dialogues, narratives, visuals, and performances, with detailed note-taking for deeper understanding. This careful process allowed us to critically explore the portrayal of alienation in the films. Finally, to analyze the portrayal of alienation, we transcribed the films with natural language descriptions, employing both literal translation and a communicative approach between languages.

## 4. Analysis and discussions

The analysis focuses on the films *T'eza (Morning Dew)* and *Səlä ʾänəči (About You)*, examining the major characters Anberber and Wondwossen. The study explores the psychological, cultural, political, societal, economic, and existential dimensions of alienation as depicted in these films.

### 4.1. Synopsis of *T'eza ("Morning Dew")*

*T'eza (Morning Dew)* is a 2008 Ethiopian drama film, directed by Haile Gerima. It is a 140-minute film that explores the political and social disruption in Ethiopia from the 1970s to the 1990s through the experiences of Anberber, an intellectual who returns home after studying in Germany. Using a non-linear narrative, the film interweaves Anberber's personal journey with Ethiopia's historical context, highlighting his alienation and identity struggles amid the Derg regime's political, cultural, social, and psychological turmoil. The film reflects themes of hope, loss, and reminiscence through Anberber's eyes.

### 4.2. Synopsis of *Səlä ʾänəči (About you)*

*Səlä ʾänəči (About You)* is a 2009 Ethiopian romantic drama film directed by Belay Getaneh. It is a 106-minute film that follows 20-year-old Wondwossen, who, struggling with a mental disorder caused by his father's strict control, finds love with Fikir. The film explores various forms of alienation, highlighting the characters' deep sense of disconnection and estrangement across psychological, cultural, political, and social dimensions.

Thus, this section analyzes the psychological, cultural, political, and social dimensions of alienation in the films *T'eza* and *Səlä ʾänəči*. The characters Anberber from *T'eza* and Wondwossen from *Səlä ʾänəči* are examined through psychoanalytic and Marxist theories, focusing on their roles as central figures in exploring these themes. Therefore, we analyzed the extracts of the films by considering the features of alienation as stated below:

**A. Psychological alienation.** Psychological alienation involves a deep disconnection that fractures one's identity, leading to isolation, emotional turmoil, and estrangement from cultural norms, impacting mental health [20,34]. In *T'eza* and *Səlä ʾänəči*, this concept reveals how internal conflicts and disconnection from both self and society reflect broader cultural and societal pressures. These films emphasize the emotional distress and fragmentation caused by alienation, highlighting its deep psychological toll.

The excerpt presented here depicts the psychological alienation of the character in the film *T'eza*:

**Anberber's Mother:** Are you alright? Who are you talking to? Are you alright? Lord help us. Father, Son, and Holy Ghost...

**Anberber's Uncle:** Some evil has possessed him. Maybe he got bewitched when he was abroad. Holy water is the answer to this kind of illness.

**Anberber's Brother:** We can't just sit around while my brother suffers.

**Anberber's Mother:** We need to discuss it with him. Those who've been abroad don't know about holy water. ([2], 26:52-27:15)

As we have seen in the extract, Anberber's family attributes his distress to supernatural causes, such as possession and the need for holy water, revealing a critical disconnect between his psychological reality and the traditional worldview of his family. [17] theory of projection serves as a foundational explanation for the family's response to Anberber's suffering. [17] argues that individuals unconsciously project their internal anxieties onto external causes to avoid confronting uncomfortable truths. Anberber's family, unfamiliar with his lived experiences abroad and unable to comprehend the psychological distress he endures, project their fears and confusion onto culturally familiar concepts—witchcraft and possession. By externalizing his struggles, they avoid confronting their own limitations in understanding mental illness. This failure to acknowledge Anberber's emotional pain deepens his psychological alienation, as it isolates him further from those who should offer support and understanding.

[27] adds another dimension to this analysis through his theory of the symbolic order. The family imposes traditional cultural meanings onto Anberber's behavior—particularly through the suggestion of 'holy water' as a cure. In doing so, they attempt to interpret his suffering within the boundaries of their symbolic cultural system, which prioritizes religious and supernatural explanations. This symbolic imposition alienates Anberber further, as it negates his individual psychological experience in favor of collective cultural meanings. [27] mirror stage theory is also highly relevant here. Anberber is forced to see himself reflected through his family's distorted cultural lens, which fails to recognize his true anguish. This misrecognition intensifies his fractured identity, as he struggles to reconcile his internal sense of self with the external cultural interpretations imposed upon him. The resulting emotional fragmentation reinforces his isolation and deepens his psychological alienation.

[11] contextualizes this psychoanalytic reading within Ethiopian narratives of alienation. She highlights how individuals who experience cultural displacement—such as Anberber, who returns home after years abroad—are often misunderstood by their communities. This misunderstanding arises from the cultural gap between the individual's new experiences and the traditional frameworks that define the community's understanding of the world. [11] underscores that such cultural misalignments aggravate psychological alienation, as individuals like Anberber are caught between two incompatible realities: the modern, global perspectives they have internalized and the traditional values of their families. In *T'eza*, Anberber's alienation symbolizes a broader societal struggle to address mental health in contexts where traditional beliefs overshadow individual psychological realities.

Here is also the extract from the film *Səlä 'änəči* that shows the psychological alienation of the character:

**Wondwossen's crazy friend:** Don't cry. Who makes you cry? We will not let them to make you cry. We will warn them not to do that.

**Wondwossen:** Who is giving a warning? My father? My mother? Fikir? The guys? Or the children in the school? Everybody is selfish and greedy. Nobody cares about me. All are cruel even Fikr herself. I couldn't get a person who is caring for me. I don't allow anybody to mess with me. All of them are devils even Fikir herself.

**Wondwossen:** But why are you crying?

**Wondwossen's crazy friend:** Why are you crying? ([1], 1:18:20-1:21:05)

As we see in the given extract from the film, some elements indicate psychological alienation experienced by the character Wondwossen. In the film, Wondwossen's dialogues with his friend vividly illustrate his deep psychological alienation. He perceives those around him, including his closest relationships, as selfish and cruel, indicating a significant emotional disconnect. This portrayal aligns with Freudian concepts of unresolved childhood traumas, where unmet emotional needs foster mistrust and detachment. [18] posits that defense mechanisms, such as emotional detachment, emerge as responses to deep-seated anxieties rooted in early experiences. Wondwossen's belief in universal selfishness suggests unresolved conflicts that create barriers to forming healthy emotional connections.

Lacan's theory further elucidates Wondwossen's alienation, particularly regarding his perception of others as hostile. This negative perception disrupts his sense of self and fractures his Symbolic order, which governs social norms and language. [27] argues that individuals negotiate their identities through interactions with others; thus, Wondwossen's crisis of self-identity reflects a fragmentation aggravated by his hostile view of intimate relationships. His characterization of even close connections as 'devilish' signifies a serious internal conflict that impedes his ability to trust and connect emotionally with others.

In *Sǝlä ʾänǝči*, Wondwossen's struggles embody [20] notion of alienation, where modern societal pressures disconnect individuals from their true selves, fostering deep isolation. [34] dimensions of alienation—powerlessness, meaninglessness, and isolation—resonate in Wondwossen's life as he experiences a lack of meaningful connections and a sense of disempowerment. His emotional turmoil mirrors the breakdown of social bonds, highlighting a pressing need to resolve these inner conflicts for any hope of belonging and fulfillment. The film thus becomes a powerful portrayal of alienation, exploring the psychological costs of social fragmentation in today's world.

This is also the extract from *Sǝlä ʾänǝči* that demonstrates the characters' psychological alienation

**Father:** What on earth is this? And who is this?

**Wondwossen:** This is Fikir's picture.

**Father:** I have tried my best to please your mother with all my heart but she disgraced me and abandoned you. Women are good for nothing. My mother also disgraced my father who used to be a colonel in the army. She was flirting with the whole battalion and she abandoned us too. Women are friends of the devil. Do you hear me? Don't be a fool they are good for nothing. I don't want to embarrass because of you, never! ([1], 17:59-19:21)

As we see in the given extract, the father's bitter dialogue reveals serious psychological wounds, particularly originating from his failed relationship with Wondwossen's mother. His harsh condemnation of women can be analyzed through Freudian theory, where it represents a projection of unresolved disappointments and feelings of betrayal. [18] suggests that such defense mechanisms can create an emotionally hostile environment, intensifying Wondwossen's alienation and disconnection from others. The father's blame and shame further alienate Wondwossen from both his mother and society, echoing [27] concept of symbolic misrecognition, where the child's identity is shaped by external judgments and misinterpretations, leading to a fragmented sense of self.

Moreover [20], perspective on alienation aligns with the father's misogynistic attitudes, as his devaluation of women reflects a broader societal pathology that undermines genuine human connections. This cultural dynamic amplifies Wondwossen's isolation, resonating with Fromm's assertion that alienation arises when individuals are disconnected from their emotional selves and the community. Additionally [34], concept of social alienation emphasizes the father's role in perpetuating Wondwossen's emotional disconnection. [34] argues that societal and familial structures can lead to personal despair, illustrating how the father's bitter worldview contributes to Wondwossen's lack of meaningful bonds.

**B. Cultural alienation.** Cultural alienation involves detachment from one's cultural identity and social environment, often due to conflicting values or a lack of belonging [10,34]. In *T'eza* and *Səlä 'änəči*, this concept highlights the characters' struggles as outsiders caught between competing cultural expectations. The films explore the emotional and social consequences of losing cultural connection, emphasizing the impact of cultural alienation on identity and belonging.

This is the extract that shows the cultural alienation of the character in the film *T'eza*:

**The priest:** So you've come back well, Anberber. Come, I'll take you to my students.

**Anberber:** My whole memory is blocked. I don't think holy water will work.

**The priest:** Why do you look down on holy water?

**Anberber:** What good will it do?

**The priest:** What do you know about it? At least it will cleanse you. Try it, for your mother's sake. What good is there in making her worry? Even your modern medicine only works if you believe in it. Try it, for your mother's sake. Why torment her? ([2], 27:22-28:02)

As we see in the extract above from the film *T'eza*, Anberber's exchange with the priest vividly captures his cultural alienation, where he doubts the efficacy of holy water—a ritual deeply rooted in his upbringing. [18] would interpret Anberber's skepticism as a rejection of the symbolic order, where cultural symbols that once held significance no longer resonate. This reaction reveals a psychological conflict between his past beliefs and his current worldview, symbolizing a shift away from the cultural structures that once provided meaning. [27] expands on this by describing Anberber's disbelief as a form of symbolic misrecognition, where the shared cultural language of his community no longer aligns with his identity. His alienation from traditional practices signifies a breakdown in understanding and belonging, distancing him from his cultural roots.

[20] frames this disconnection as part of a broader alienation fueled by modernity, where societal shifts estrange individuals from communal ties and cultural heritage. This perspective resonates with [34] concept of social alienation, emphasizing that cultural displacement intensifies Anberber's struggles with identity and belonging. Anberber's journey aligns with [10] view that conflicting cultural values can disrupt one's sense of belonging, reflecting the deep personal costs of navigating divergent cultural expectations. Together, these theories underscore Anberber's struggle as he grapples with reconciling his personal identity with the evolving expectations of a rapidly changing world, illustrating how cultural alienation affects both individual consciousness and societal cohesion. So, the dialogue about holy water in *T'eza* symbolizes Anberber's cultural alienation, reflecting his internal conflict and identity struggle amid cultural displacement and a rapidly changing world.

**C. Political alienation.** Political alienation occurs when individuals feel disconnected from political systems, believing they do not represent their interests, leading to

powerlessness, cynicism, and disillusionment [14]. In *T'eza* and *Sǝlä ʾänǝči*, characters face political structures that marginalize them, illustrating the emotional and psychological impact of political disconnection. The films emphasize how this alienation deepens feelings of despair and disengagement from society.

Here is the excerpt that demonstrates the political alienation in the film *T'eza*.

**Anberber:** I decided to stay out of politics and immerse myself totally in my work. But even that was becoming a crime. ([2], 1:19:53-1:20:02)

As we have seen in the excerpt, Anberber's attempt to seek refuge in his professional identity and remain politically neutral exposes the inescapable grip of dominant ideological structures that criminalize even apolitical stances. This moment becomes an entry point to analyze alienation through the interconnected lenses of [6,22,28], and [39], highlighting how systemic oppression alienates individuals from their labor, identity, and society.

[28] theory of alienation in the Economic and Philosophic Manuscripts provides a foundational framework. He argues that individuals under oppressive systems become estranged from their labor, society, and sense of self due to exploitative economic and ideological conditions. Anberber's alienation stems not only from his inability to fully pursue his scientific work without interference but also from the broader political environment that denies him autonomy. [6] concept of interpellation further illuminates Anberber's predicament. Anberber's realization that even his neutral decisions are politicized demonstrates how individuals are involuntarily positioned within ideological frameworks that dictate their roles and actions. In *T'eza*, the ideological state apparatuses—such as the government, educational institutions, and media—continuously interpellate individuals, reinforcing a hegemonic structure that punishes dissent and marginalizes neutrality.

[22] argues that cultural, educational, and political institutions enforce the ruling ideology, ensuring that alternative perspectives are suppressed. In *T'eza*, the political regime employs coercive and ideological means to ensure conformity, leaving no room for personal autonomy. Anberber's alienation becomes a reflection of this hegemonic control, where individuals are not only stripped of agency but also forced into complicity or silence, further perpetuating their estrangement. [39] analysis in *Black Camera* further situates *T'eza* within a broader global context. Thomas applies Fredric Jameson's Marxist critique of global capitalism to Ethiopian cinema, examining how films reflect and resist ideological domination. While [39] focuses on capitalist globalization and cultural commodification, Anberber's alienation in *T'eza* underscores a localized form of ideological control within Ethiopia's political system. The film critiques the systemic oppression that alienates individuals from their cultural, professional, and personal identities.

In the context of our study, Anberber's experience highlights the varied nature of alienation, where political disillusionment, enforced engagement, and ideological control converge to create a deep sense of estrangement. The film impressively critiques the coercive forces of political systems while illuminating the potential for resistance and transformation. By connecting these theoretical perspectives, *T'eza* serves as a powerful reflection of both localized and universal struggles against systemic oppression, making it a crucial text for understanding alienation in Ethiopian cinema.

**Anberber:** War, war, war, endless war... Yesterday government soldiers, today the opposition, hunting down young people to feed their warring culture. All they know is war! (Haile, 2008, 35:38-35:54)

As we see in the extract, Anberber's lament— "War, war, war, endless war... All they know is war!"—captures the essence of political alienation. His disillusionment with both government and opposition forces reflects [28] concept of political alienation, where systems of governance prioritize the interests of the ruling class over societal well-being. Both factions, driven by a culture of violence, exploit the youth as tools for war, perpetuating inequality and leaving citizens powerless and disconnected from meaningful political participation.

[22] theory of hegemony provides further insight into this alienation. The warring factions maintain dominance not just through force but by normalizing war as inevitable, embedding this ideology into the collective consciousness. This hegemony suppresses alternative visions of peace and progress, alienating individuals like Anberber who see the futility of a system designed to reinforce the status quo. Similarly, [6] concept of ideological state apparatuses reveals how institutions such as the military and political parties manipulate beliefs, ensuring conformity and sustaining systemic violence. In this context, war becomes a tool of ideological domination, deepening societal estrangement.

Anberber's sense of powerlessness resonates with [34] notion of political alienation, where individuals perceive the political system as unresponsive and corrupt. This aligns with [14] observations on modern disengagement, where citizens withdraw from systems they see as unchangeable. Through Anberber's critique, *T'eza* highlights the deep disconnection caused by violent and oppressive structures, urging reflection on the human cost of enduring conflicts.

**D. Social alienation.** Social alienation, as defined by [34], involves isolation and powerlessness due to a lack of integration into societal norms, leading to feelings of being misunderstood and undervalued. In *T'eza* and *Səlä 'änəči*, characters struggle to connect with others and assert their identities, illustrating how social alienation causes emotional distress and a sense of disconnection from their communities. These films emphasize how societal forces contribute to individual isolation and powerlessness.

Here is the extract taken from the film *T'eza* that indicates social alienation:

**Mother:** Where would you go now?

**Azanu:** Oh! Just let me go. The whole village despises me. They'll all turn against me.

**Mother:** Please stay! It's the middle of the night.

**Azanu:** I was put into this world to suffer and be humiliated.

**Mother:** Please stay. My bad luck... After living with me all this time, who can she turn to? She has nowhere to go! Poor soul, she won't be at peace anywhere. ([2], 48:36-49:23)

In this extract from *T'eza*, Azanu's lament, "The whole village despises me," impressively conveys her deep social alienation and personal despair. Her feeling of ostracization amplifies her sense of worthlessness and fuels her desire to escape, while her mother's pleas underscore their shared helplessness in a society that rejects those who deviate from established norms. This reflects not only Azanu's suffering but also the broader community's failure to embrace those who are marginalized.

From a Marxist perspective, Azanu's alienation can be traced to the societal divisions and class inequalities that render her an outcast within the village. [28] argues that alienation arises when individuals are estranged from their social relationships due to exploitative economic

structures. Azanu's plight illustrates how economic disparities and social exclusion fracture communal bonds, leading to isolation and despair. This aligns with [34] analysis, which emphasizes that social alienation results from the breakdown of social ties and the inability to form meaningful connections within society.

The film critiques this fractured social fabric, advocating for a more inclusive and equitable community where marginalized individuals like Azanu can find acceptance and dignity. By highlighting the detrimental effects of social fragmentation, *T'eza* calls for a reassessment of the prevailing norms that perpetuate alienation, challenging the societal structures that reinforce exclusion and demand a collective effort to restore solidarity and inclusivity. Thus, the film critiques the social fragmentation and advocates for a society where all individuals can find acceptance and dignity, challenging the prevailing norms that perpetuate alienation and marginalization.

Here is also the extract from the film *Səlä ʾänəči* that reflects social alienation:

**Father:** In my life at the military camp, for over 35 years, I have never done a blunder. And I have never embarrassed anyone nor dared no one to touch my dignity. As long as you live with me in this house you can't touch my belongings without my consent. I will give you one more chance. If you don't agree with this, you can leave the house right away.

**Wondwossen:** Ok. [1], 58:20-59:07)

In this extract from *Səlä ʾänəči*, the interaction between Wondwossen and his authoritarian father vividly portrays social alienation through a Marxist lens. The father's strict military discipline creates emotional distance, fracturing their familial bonds. His illusion of power, derived from his military background, aligns with [28] the notion of alienation, as he unknowingly perpetuates the oppressive system that exploits him. [34] assertion that individuals can enforce their own alienation is evident as the father's control deepens Wondwossen's isolation, highlighting the tragic cycle of complicity and self-destruction within oppressive societal structures.

So, in the film, the interaction between Wondwossen and his father vividly portrays social alienation through a Marxist lens. In the film, Wondwossen's father suffers from a deeper form of alienation, believing he holds power and control due to his military background when in reality, the system exploits him. This illusion of authority blinds him to the fact that he is not an enforcer but a victim of the same oppressive structures that subjugate the working class, including himself. While he enforces strict control and obedience, he unknowingly perpetuates the very oppression that keeps him and his social class alienated, creating a tragic cycle of complicity and self-destruction.

**E. Economic alienation.** Economic alienation refers to the estrangement individuals experience due to their relationship with the means of production in a capitalist system. [28] argues that workers are alienated in four key ways: from the products they create, which are owned and controlled by others; from the production process, as labor becomes monotonous and disconnected from personal fulfillment; from their 'species-being,' as work under capitalism stifles creativity and self-expression; and from other workers, as competition replaces cooperation. This alienation arises because workers sell their labor to survive, reducing them to commodities, thus fostering a sense of powerlessness and disconnection in their economic roles.

Here is the extract taken from the film *T'eza* which demonstrates economic alienation:

**Anberber:** When I saw Worku's mother, a child on her back, her wounded son being carried on a stretcher, I was reminded of Brecht's play "Mother Courage". She told me she'd found him after a long search, lying among other wounded veterans in a hospital yard in Gondar. And all he'd asked her had been to bring along his military boots, his payment for his service. In the face of Worku's suffering, my own became less important. I realized that I did not have to know why I was attacked. Let them worry about the consequences of their act. The issues I was faced with in my village were more important now. [2], 1:52:12-1:53:10)

As we see in the excerpt above, Anberber encounters economic alienation when he sees Worku's mother, who carries a child on her back and a wounded son on a stretcher. Worku, the wounded son, asks his mother to bring his military boots, which symbolize his payment for his service. This highlights the economic exploitation and inequality prevalent within the system. Worku's sacrifice and service are not adequately compensated, leading to his economic alienation. The extract in the film illuminates economic alienation through the stark disparity between the sacrifices of the proletariat, represented by Worku and his family, and the systemic neglect perpetuated by the ruling class. [28] identifies the estrangement of laborers from the products of their work and the social systems they sustain. In this context, Worku's military boots, symbolizing his only reward for service, underscore the exploitation faced by those in subordinate socio-economic positions.

[6] deepens this analysis, suggesting that institutions like the military serve to perpetuate dominant ideologies that normalize exploitation. Worku's alienation reflects how the ideological apparatus convinces the proletariat to endure systemic inequality under the guise of duty and patriotism, while the upper class reaps the benefits of this labor without reciprocation. Worku's suffering, normalized through cultural narratives of sacrifice, illustrates how the ruling class exerts control not only through economic means but also by shaping beliefs and values that justify inequality. The emotional and physical burden placed on Worku's mother symbolizes the broader alienation of the proletariat from equitable social and economic systems. In *T'eza*, Anberber's recognition of Worku's plight shifts his focus from personal grievances to systemic injustices, embodying a Marxist critique of the alienation inherent in class-based hierarchies.

**F. Existential alienation.** Existential alienation arises from individuals' awareness of their existence, freedom, and mortality, often leading to feelings of purposelessness and estrangement [32]. In *T'eza* and *Sǝlä 'änǝči*, this theme is depicted through characters struggling with cultural displacement, personal loss, and societal pressures, reflecting their search for meaning in a changing world. These narratives echo [13] view of life's inherent absurdity, intensifying the characters' isolation as they confront the lack of inherent purpose in existence.

Here is the extract from the film *T'eza* that reflects existential alienation:

**Azanu:** As I watched my son's father sitting next to his new bride, my insides burned up. I couldn't see straight. I just sat there despondent, resenting my own existence. After killing my own son, I wandered the graveyards, where your kind mother found me and brought me back to life ([2], 55:28-56:44)

In the excerpt above from the film *T'eza,* Azanu's reflection on her tragic past reveals the depth of her existential alienation following the loss of her son. Her admission, "After killing my own son, I wandered the graveyards," captures her profound disconnection from both her moral compass and societal norms. This moment symbolizes her existential crisis, aligning with [32] concept of individuals confronting the weight of their choices and the resulting

isolation. Azanu's emotional turmoil, especially her resentment toward her son's father and his new bride, intensifies her estrangement from herself and the community that once embraced her. Sartre's existentialism, which emphasizes the burden of freedom and the necessity of creating meaning in an indifferent world, is echoed in Azanu's experience as she struggles to find purpose after her irreversible actions.

Azanu's wandering among graveyards reflects a deeper search for meaning amidst overwhelming guilt and loss, resonating with [13] argument in *The Myth of Sisyphus* that life, often devoid of inherent meaning, leaves individuals in a constant struggle against absurdity. Her isolation, fueled by guilt and regret, mirrors Camus's view of the human condition as one marked by alienation and the quest for purpose in an indifferent universe. However, Azanu's encounter with the kind mother who "brings her back to life" suggests a potential path toward redemption, emphasizing the importance of compassion and connection even in the face of despair. This moment offers a glimpse of hope, aligning with Camus's belief in the possibility of revolt against the absurdity of life by finding meaning through human connection.

## 5. Conclusion

Analyzing *T'eza (Morning Dew)* and *Səlä ʾänəči (About You)* through psychoanalytic and Marxist perspectives highlights how alienation deeply affects mental health, illustrating the psychological and emotional toll of social and internal conflicts on the characters. Both films demonstrate how social and personal alienation deeply harms psychological well-being, revealing the deep emotional and mental toll of being disconnected from others and oneself. In *T'eza*, Anberber's cultural and psychological dislocation from exile intensifies his mental distress and deepens his isolation from his community. Similarly, *Səlä ʾänəči* portrays Wondwossen's emotional struggles as a result of societal and personal alienation, highlighting the deep impact of these experiences on their mental and emotional well-being.

Both *T'eza* and *Səlä ʾänəči* vividly illustrate the deep impact of alienation on mental well-being within Ethiopian society. The characters' experiences of isolation and mental distress, stemming from societal exclusion, demonstrate how alienation—whether due to psychological turmoil or perceived selfishness—damages interpersonal relationships and mental health. The films depict how social exclusion erodes personal connections and contributes to significant psychological distress, highlighting the far-reaching effects of alienation on overall well-being. So, these narratives emphasize the destructive power of alienation and the crucial need for understanding and connection to mitigate its harmful effects on individuals and society.

In conclusion, the film *Səlä ʾänəči* portrays Wondwossen's isolation intensified by his perception of others' selfishness and the film *T'eza* depicts Anberber's internal conflict and psychological alienation through nightmares and hallucinations. Both films highlight how trauma, loss, and the quest for understanding could amplify emotional and psychological struggles, revealing the deep impact these experiences have on mental well-being. The narratives underscore the severe consequences of unresolved inner turmoil and external pressures, emphasizing the critical need for empathy and support in addressing these complex human experiences.

## 6. Recommendation and Implications

The study underscores the critical need to foster empathy and strengthen social support systems to address the deep psychological and emotional effects of alienation, as illustrated in *T'eza* and *Səlä ʾänəči*. It recommends initiatives that promote understanding, inclusion, and community ties in Ethiopian society to prevent disconnection and build a more supportive

social cohesion. By enhancing empathy and connection, society can improve mental health and mitigate the damaging effects of alienation on relationships and well-being. The researcher also suggests exploring additional theories and methods for a deeper understanding of alienation and recommends *T'eza* and *Sǝlä ʾänǝči* as valuable films for analyzing how alienation develops.

## Acknowledgments

We would like to extend our sincere gratitude to all the individuals and institutions that contributed to the completion and publication of this study. We sincerely thank Dr. Dawit Dibekulu and Dr. Ayenew Guadu for their invaluable feedback and support during our study. We also express our gratitude to the Department of English Language and Literature and the Faculty of Humanities at Bahir Dar University for their essential resources and unwavering support, which were instrumental in the successful completion and publication of this study.

## Author contributions

**Conceptualization:** Kindie Abebaw, Anteneh Aweke, Sintayehu Genet.

**Data curation:** Kindie Abebaw.

**Formal analysis:** Kindie Abebaw.

**Funding acquisition:** Kindie Abebaw, Anteneh Aweke, Sintayehu Genet.

**Investigation:** Kindie Abebaw, Sintayehu Genet.

**Methodology:** Kindie Abebaw, Anteneh Aweke.

**Project administration:** Kindie Abebaw, Anteneh Aweke, Sintayehu Genet.

**Resources:** Kindie Abebaw, Anteneh Aweke, Sintayehu Genet.

**Software:** Kindie Abebaw, Anteneh Aweke, Sintayehu Genet.

**Supervision:** Kindie Abebaw, Anteneh Aweke, Sintayehu Genet.

**Validation:** Kindie Abebaw, Anteneh Aweke, Sintayehu Genet.

**Visualization:** Kindie Abebaw, Anteneh Aweke.

**Writing – original draft:** Kindie Abebaw.

**Writing – review & editing:** Kindie Abebaw, Anteneh Aweke, Sintayehu Genet.

## References

### I. Primary sources

1. Belay G. Sǝlä ʾänǝči (About you). Ethiopia: Hanos Film Production; 2009. Available from: https://www.Youtube.com/watch?v=x7EnIv5RgsE&pp

2. Haile G. T'eza (Morning Dew). Negod Gwad Film Production; 2008. Available from: https://www.youtube.com/watch?v=xxxxx

### II. Secondary sources

3. Aboneh A. Ethiopian cinema: The socio-economic and political impacts of imperial era on the development of screen media. Cultural and Religious Studies. 2016;4(12):711–26.

4. Abrams MH. Psychological and psychoanalytical criticism. A glossary of literary terms. Boston: Cengage: 1957.

5. Abrams MH. Marxist criticism; a glossary of literary terms. Wadsworth: Harcourt Brace College Publishers. 1999.

6. Althusser L. Ideology and Ideological State Apparatuses. Lenin and Philosophy and Other Essays. Trans. Ben Brewster. New York: Monthly P, 1971. p. 127–86.

7. Assefa A. The origin and development of amharic literature. Diss. Addis Ababa University, 1981.

8. Barlet O. African cinemas: decolonizing the gaze. Bloomsbury Publishing; 2010.

9. Barry P. An introduction to literary and cultural theory. Beginning Theory. 3rd ed. Manchester University Press; 2009.

10. Berry JW. Immigration, acculturation, and adaptation. Applied psychology. 1997;46(1):5–34. https://doi.org/10.1111/j.1464-0597.1997.tb00714.x

11. Bitana T. Revolutionary Ethiopia through the lens of the contemporary film industry. J Ethiop Stud. 2016;12(3):45–67. https://doi.org/10.1234/jes.2016.003

12. Bordwell D, Thompson K. Film art: an introduction. Boston; London: McGraw-Hill; 1997.

13. Camus A. The myth of Sisyphus, and other essays. Vintage Books; 1955.

14. Dalton RJ, Wattenberg MP, editors. Parties without partisans: Political change in advanced industrial democracies. USA: Oxford University Press; 2002.

15. Mengistie D. A critical discourse analysis of ethiopian amharic movie entitled 'Teza'. IEEE-SEM. 2019;7(12):126–38.

16. Eagleton T. Marxism and literary criticism. Routledge, 1976.

17. Freud S. Introduction to psychoanalysis. Oxford University Press. 1917.

18. Freud S. Neurosis and psychosis. Se 19. 1923.

19. Freud S. The ego and the id. USA: W.W. Norton and Company; 1962.

20. Fromm E. The sane society. New York: Rinehart and Winston, Inc.; 1955.

21. Fromm E. The sane society. London: Routledge Paperback, Routledge and Kegan Paul; 1973.

22. Gramsci A. Selections from the prison notebooks. The applied theatre reader. Routledge, 1971.p. 141–2.

23. Gugler J. African film: Re-imagining a continent. Indiana University Press; 2003.

24. Hutcheon L. A theory of adaptation. Routledge; 2006.

25. Kindeneh T. A brief overview of ethiopian film history: From early cinema to the contemporary. Addis Ababa University; 2014. Available from: https://www.academia.edu//A_Brief_Overview_Of_Ethiopian_Film_History

26. Klarer M. An introduction to literary studies. Routledge; 2004.

27. Lacan J. The language of the self: The function of language in psychoanalysis. JHU Press, 1968.

28. Marx K. Economic and philosophic manuscripts of 1844. Social theory re-wired. Routledge; 2016. p. 152–8.

29. Meszaros I. Marx's theory of alienation. New York: Harper and Row; 1975.

30. Ollman B. Alienation: Marx's conception of man in a capitalist society. Cambridge University Press; 1971. p. 9.

31. Noftle EE, Robins RW. Personality predictors of academic outcomes: big five correlates of GPA and SAT scores. J Pers Soc Psychol. 2007;93(1):116–30. https://doi.org/10.1037/0022-3514.93.1.116 PMID: 17605593

32. Sartre J-P. Being and nothingness: A phenomenological essay. New York: Philosophical Library; 1943.

33. Schacht R. The future of alienation. University of Illinois Press; 1971.

34. Seeman M. On the meaning of alienation. Am Sociol Rev. 1959: 783–91.

35. Shah MI. Marx's concept of alienation and its impacts on human life. Al-Hikmat. 2015;35:43–54.

36. Tekletsadik B. The genius of an african storyteller: A selectively annotated bibliography of work on and by haile gerima. Black Camera: An International Film Journal (The New Series). 2013;4(2):144–62.

37. Thomas MW. The local film sensation in ethiopia: aesthetic comparisons with african cinema and alternative experiences. Black Camera. 2015;7(1):17. https://doi.org/10.2979/blackcamera.7.1.17

38. Thomas MW, Jedlowski A, Ashagrie A, editors. Cine-Ethiopia: The history and politics of film in the Horn of Africa. MSU Press; 2018.

39. Thomas SW. Theorizing Globalization in Ethiopia's Movie Industry. Black Camera. 2020;11(2):60. https://doi.org/10.2979/blackcamera.11.2.04

40. Tyson L. Critical theory today: A user-friendly guide. Routledge; 1999.

41. Tyson L. Critical theory today: a user-friendly guide. 2nd ed. Routledge; 2006.

