## [Decision Letter · Decision Letter 0]

23 Sep 2024

PONE-D-24-38471Alienation in Ethiopian Cinema: "T'eza" ("Morning Dew") and "Səlä ʾänəči"("About you") in FocusPLOS ONE

Dear Dr. Mitiku,

Thank you for submitting your manuscript to PLOS ONE. After careful consideration, we feel that it has merit but does not fully meet PLOS ONE’s publication criteria as it currently stands. Therefore, we invite you to submit a revised version of the manuscript that addresses the points raised during the review process. In overall, the reviewers were positive about your manuscript. Reviewer 1 has added an alternate perspective on one of the characters analyzed. I recommend adding their opinion, either in the text or as a footnote. Reviewer 2, however, had more issues with the manuscript. They list five points that require detailed answers, and they might involve rewriting large sections of the manuscript - which requires to be written as a proper academic article, in order to publishable as part of a journal, instead of a dissertation.

We look forward to receiving your revised manuscript.

Kind regards,

Rafael Galvão de Almeida, PhD.

Academic Editor

PLOS ONE

3. We note that you have referenced (Minasie Gessesse (2010). A semiotic critical discourse analysis of selected Amharic films of Ethiopia: With reference to Zuma and Semayawi Feres) which has currently not yet been accepted for publication. Please remove this from your References and amend this to state in the body of your manuscript: (ie “Bewick et al. [Unpublished]”) as detailed online in our guide for authors http://journals.plos.org/plosone/s/submission-guidelines#loc-reference-style

Reviewers' comments:

Reviewer's Responses to Questions

**Comments to the Author**

1. Is the manuscript technically sound, and do the data support the conclusions?

Reviewer #1: Yes

Reviewer #2: Partly

2. Has the statistical analysis been performed appropriately and rigorously? 

Reviewer #1: N/A

Reviewer #2: N/A

3. Have the authors made all data underlying the findings in their manuscript fully available?

Reviewer #1: No

Reviewer #2: Yes

4. Is the manuscript presented in an intelligible fashion and written in standard English?

Reviewer #1: Yes

Reviewer #2: Yes

5. Review Comments to the Author

Reviewer #1: The work was conducted within the journal's publishing rules, analyzing data with good criteria that's supported by solid literature evidence. Although, there's two observations: 1) in the page 23 the author makes an analysis of the social alienation on the film Səlä ʾänəči, he claims that "The father's strict control, shaped by his military background, symbolizes the bourgeoisie’s dominance over the proletariat, mirroring broader societal class struggles". From my perspective, the father - as well as the mlitary forces of the country itself - is also a working class victim of the opressive system that runs control over people as much as the son, but he suffers from a different kind of alienation: the one that instrumentalizes people who are not from the upper classes of the society, but are seduced and blinded with a sense of power and control used, at the end, for the oppression of himself and the ones of his kind (view indications of literature bellow). 2) The link for the the film T'eza is missing a part of it.

Indicative literature: Horkheimer and Adorno's Dialetics of Enlightainment: 2002, pages 137-172, "Elements of Anti-Semitism": https://monoskop.org/images/2/27/Horkheimer_Max_Adorno_Theodor_W_Dialectic_of_Enlightenment_Philosophical_Fragments.pdf

Theodor Adorno's "The authoritan personality": https://ia601506.us.archive.org/28/items/THEAUTHORITARIANPERSONALITY.Adorno/THE%20AUTHORITARIAN%20PERSONALITY.%20-Adorno.pdf

Reviewer #2: The subject of Ethiopian cinema studies is a relatively new and growing subfield within the field of African film studies, so the essay under review – Alienation in Ethiopian Cinema – is a welcome addition. The essay is somewhat ambitious in trying to use both Marxist and Freudian theory to analyze two movies: Teza by the most famous Ethiopian director Haile Gerima and Selanchi by a less-known but nevertheless important director Belay Getaneh. One might have cautioned the authors to limit their analysis to either Freud or Marx, rather than both of them, but the authors aim to show that there are different theoretical approaches to thinking about “alienation” as it is expressed in films that lead to different conclusions.

I think the authors have begun a worthwhile project, but there are several problems with essay that would have to be resolved before publication. I hope the authors will continue working on it, because I think their argument does have potential, so I would be willing to read a significantly revised draft. I have a lot of comments here, and I don’t intend all of them as criticism, but as genuine conversation and excitement about the topic and its potential to eventually become a good essay.

FIRST, one problem is the scholarly conversation. The essay claims that it offers a uniquely Marxist and psychoanalytic approach to Ethiopian cinema, but it seems unaware of other scholarship that is also interested in that subject. For example, Steven Thomas’s article (2020) “Globalization in Ethiopia’s Movie Industry” in the journal Black Camera is informed by Marxist theories of globalization such as Fredric Jameson to analyze popular Ethiopian movies such as Rebuni and YeWendoch Guday. In addition, Steven Thomas’s chapter on Ethiopian cinema in the second, updated edition of African Film Studies (Routledge 2023) compares the movie Teza with another film by Getaneh Belay. There is a special 2013 issue of the journal Black Camera devoted to the movie Teza that includes an analysis of that movie through the lens of Frantz Fanon (whose theories of alienation were informed by both Marx and psychoanalysis.) That special issue also includes a useful annotated bibliography on Haile Gerima put together by the Ethiopian scholar Tekletsadik Belachew. Moreover, Bitania Tadessa’s essay “Revolutionary Ethiopia through the Lens of the Contemporary Film Industry” (2016) uses psychoanalytic methods to analyze several Ethiopian films, including Teza. Finally, there is a collection of essays on Ethiopian cinema titled Cine-Ethiopia: the History of Film and Politics in the Horn of Africa, edited by Aboneh Ashagrie, Michael Thomas, and Alessandro Jedlowski (Michigan State University Press 2018) that would certainly be useful. So, it’s strange that none of these sources are mentioned, since some of them are major works in the field of Ethiopian film studies.

TWO is how scholarly sources are used. In addition to acknowledging that such sources exist, the writers of the essay need to actually say what other secondary sources argue. It is not enough to say there are secondary sources. One has to engage with their arguments, their reasoning, and their use of evidence and have a genuine scholarly conversation with the other sources. The problem with this essay is that it acknowledges that other sources exist at the beginning of the essay, but then seems to forget about them, as they are never mentioned again later as they are analyzing the movies. The point should be to use these sources to deepen their analysis of the films.

THREE, stylistically, the essay reads like an MA thesis for university where a section of the essay summarizes the college textbook on theory X and theory Y, sure to mention all of the old books that were on the reading list for class. And this section is followed by a section section where the goal is to prove that theory X and theory Y are exhibited in the movies. My recommendation is that instead, the authors begin with questions about how to interpret and read the movies, and then bring in the theory and the secondary scholarship as they explore these questions.

For example, the authors note that the main character of Anberber suffers psychological trauma, which could be read as alienation from society. But the movie explores Anberber’s trauma as a much broader question. Noticeably missing from their analysis are several other traumatic events, such as his forced exile after the Derg regime killed his best friend and his getting attacked by racist thugs in Germany. One does not need Freud and Lacan to appreciate why such violence would cause him to suffer emotionally, and the point of the movie is that Anberber’s past suffering is repeating itself when he returns to his town to discover that a civil war is causing the young men to hide in the hills. Anberber clearly identifies with the young men since montage editing projects his own image in place of a boy who is shot and killed. None of this is mentioned in the essay, though arguably it is the main point of the movie. Later, at the end of the essay, there is discussion of the alienation felt by Azanu, but the essay never tells you who Azanu is, and never mentions the significant detail that she killed her own child. My point here is that the essay could do a better job of discussing details of the movie in terms of the whole story. The problem is that the essay sometimes seems to force details into proving the Freudian or Maxist point without considering how those details are part of a story with many characters and themes. Never mentioned in the analysis is how the movies begin and how they end – the stories.

Likewise, in the movie Selanchi, one could read Wondwossen’s abusive and demanding father as a classic example of the “super ego.” But one could also argue that the real superego is not just the father, because the whole community (including the school) places cruel demands on the main character. What is remarkable in several scenes is how Wondwossen tries to construct a male identity for himself (or “ego-ideal” in Freudian terms), but always fails to figure out the social norms (or “symbolic order” in Lacanian terms.) For example, there are scenes of him looking in the mirror trying to figure out how to dress himself – like Lacan’s point about the mirror stage. So, this is a matter of debate. Is the problem the father and Wondwossen’s competition for Fiker’s love and affection, Teferi? Or is it the whole society that is the problem? One might disagree that the father figure in Selanchi is a Freudian father and instead argue that he represents Ethiopia’s troubled history that still affects young people. This is an important question because it signals the difference between Freudian approaches that focus on individual family narratives and Lacanian approaches that aim to expose broad ideological and social contradictions and Marxist approaches that foreground historical dialectic. Noticeably, the character Fiker, whose name means “love”, is never discussed in the essay, nor is the fact that the mother is an absent presence throughout. Where is the love?

FOUR, theoretically, one might raise the question about how we put Marx and Freud together. There are many theorists who have synthesized Marx and Freud such as Althusser and Fanon, but the authors instead take a different tactic. They have a section using Freud to analyze the movies, and a separate section using Marx, as if Freud and Marx are different. But are they?

FIVE is the analysis of the films themselves. In terms of analyzing the films, the analysis relies entirely on excerpts of dialogue, but what about the cinematic image, point of view shots, extra-diegetic sound, narrative structure, plot, and montage editing? Moreover, if the point of the essay is to argue that these movies reflect their social and political context, what is weirdly missing from the essay is any analysis of that social and political context. The movie Teza very deliberately juxtaposes two historical moments – the moment in 1974 when Haile Selassie is overthrown and the Derg takes power and the moment in 1990 when the civil war is leading to the end of the Derg regime. Is Haile Gerima saying these two moments in history are similar? I don’t think so, but then what? What might be useful is a work of historical scholarship that uses Freud and Marx to analyze the problems of the revolution, such as John Markakis and Nega Ayele’s book Class and Revolution in Ethiopia (1978). Likewise, in the movie Selanchi, Wondwossen’s father was also clearly traumatized by his role as a soldier during the Derg regime, so how are we to understand those historical allusions in the movie? One could argue that Wondwossen represents the question of how to be a man in the new, post-Derg nation. For example, here is where they might look to Steven Thomas’s essay on the movies Rebuni and YeWendoch Guday, Michael Thomas’s analysis of the movie Siryat, and Tekletsadik Belachew’s chapter on the religious symbolism in Teza. All of these also reflect on the historical contexts of Ethiopia in the early 2000s and bring in details of that context. Do these other scholarly essays on Ethiopian movies support the argument in this essay about Teza and Selanchi? Could this essay bring in some of the cultural and political context mentioned in these other essays in support of their analysis?

6. PLOS authors have the option to publish the peer review history of their article (what does this mean? ). If published, this will include your full peer review and any attached files.

**Do you want your identity to be public for this peer review?** For information about this choice, including consent withdrawal, please see our Privacy Policy .

Reviewer #1: No

Reviewer #2: No

---

## [Author Response · Author response to Decision Letter 1]

5 Nov 2024

Dear Reviewers,

Thank you for valuable comments and suggestions on our article. It was instrumental to shape our paper. We tried to revise and edit all the comments on our paper and write in different color changes based on the reviewer code.

We are also happy to accept all comment after rereading our revised paper.

Thank you for all!

---

## [Decision Letter · Decision Letter 1]

15 Nov 2024

PONE-D-24-38471R1Alienation in Ethiopian Cinema: "T'eza" ("Morning Dew") and "Səlä ʾänəči"("About you") in FocusPLOS ONE

Dear Dr. Mitiku,

Thank you for submitting your manuscript to PLOS ONE. After careful consideration, we believe the manuscript is close from meeting PLOS ONE’s publication criteria as it currently stands. Therefore, we invite you to submit a revised version of the manuscript that addresses the points raised during the review process. All the main issues with the article have been cleared. The reviewers requested you to clarify some points, give a short contextualization of the Derg regime, if possible, relate your research to the previously published literature, and verify your reference list.

We look forward to receiving your revised manuscript.

Kind regards,

Rafael Galvão de Almeida, PhD.

Academic Editor

PLOS ONE

Journal Requirements:

Reviewers' comments:

Reviewer's Responses to Questions

**Comments to the Author**

1. If the authors have adequately addressed your comments raised in a previous round of review and you feel that this manuscript is now acceptable for publication, you may indicate that here to bypass the “Comments to the Author” section, enter your conflict of interest statement in the “Confidential to Editor” section, and submit your "Accept" recommendation.

Reviewer #2: All comments have been addressed

2. Is the manuscript technically sound, and do the data support the conclusions?

Reviewer #2: Partly

3. Has the statistical analysis been performed appropriately and rigorously? 

Reviewer #2: Yes

4. Have the authors made all data underlying the findings in their manuscript fully available?

Reviewer #2: Yes

5. Is the manuscript presented in an intelligible fashion and written in standard English?

Reviewer #2: Yes

6. Review Comments to the Author

Reviewer #2: The authors have made good improvements to the essay and incorporated relevant sources.

I think the later sections of the essay could better dialogue with the work of their colleagues Bitania Tadesse and Steven Thomas, whom they mention only at the beginning, but whose work relates to theirs significantly. Therefore, the authors might want to consider that conversation among scholars of Ethiopian cinema as being slightly more interesting to readers than the survey of old theory.

Moreover, considering Teza's focus on the Derg regime, I wonder why there is so little information about Ethiopia's political and economic history. But nevertheless, the essay stands and does what it purports to do, which is read the films through the lenses of some Marxist and Freudian theory.

The only absolutely necessary change is the references page, where some of the references are incomplete or improperly formatted according to the style guidelines. That is a problem, but it is a problem that can be easily and quickly resolved.

7. PLOS authors have the option to publish the peer review history of their article (what does this mean? ). If published, this will include your full peer review and any attached files.

**Do you want your identity to be public for this peer review?** For information about this choice, including consent withdrawal, please see our Privacy Policy .

Reviewer #2: No

---

## [Author Response · Author response to Decision Letter 2]

23 Dec 2024

We would like to express our sincere gratitude for your valuable comments, which have greatly contributed to the improvement of the paper as well as to the journal’s commitment to publishing high-quality articles. As the authors of this article, we have made every effort to address all the comments raised during the second round of review. To facilitate easy identification and cross-checking, we have highlighted the changes in color to ensure that each comment has been clearly addressed.

---

## [Editor Report · Decision Letter 2]

26 Dec 2024

Alienation in Ethiopian Cinema: "T'eza" ("Morning Dew") and "Səlä ʾänəči"("About you") in Focus

PONE-D-24-38471R2

Dear Dr. Mitiku,

We’re pleased to inform you that your manuscript has been judged scientifically suitable for publication and will be formally accepted for publication once it meets all outstanding technical requirements.

Kind regards,

Rafael Galvão de Almeida, PhD.

Academic Editor

PLOS ONE

Additional Editor Comments (optional):

Some minor edits are required, like italicizing some terms like journal and book titles. Also I can suggest changing the first sentence to "“Stories, in their many mediums (literature, film, comics, etc.)...”
---

## [Editor Report · Acceptance letter]

PONE-D-24-38471R2

PLOS ONE

Dear Dr. Abebaw,

I'm pleased to inform you that your manuscript has been deemed suitable for publication in PLOS ONE. Congratulations! Your manuscript is now being handed over to our production team.

Kind regards,

on behalf of

Dr. Rafael Galvão de Almeida

Academic Editor

PLOS ONE